# 2D material programming for 3D shaping

Amirali Nojoomi [1], Junha Jeon[2] & Kyungsuk Yum [1✉]

Two-dimensional (2D) growth-induced 3D shaping enables shape-morphing materials for diverse applications. However, quantitative design of 2D growth for arbitrary 3D shapes remains challenging. Here we show a 2D material programming approach for 3D shaping, which prints hydrogel sheets encoded with spatially controlled in-plane growth (contraction) and transforms them to programmed 3D structures. We design 2D growth for target 3D shapes via conformal flattening. We introduce the concept of cone singularities to increase the accessible space of 3D shapes. For active shape selection, we encode shape-guiding modules in growth that direct shape morphing toward target shapes among isometric configurations. Our flexible 2D printing process enables the formation of multimaterial 3D structures. We demonstrate the ability to create 3D structures with a variety of morphologies, including automobiles, batoid fish, and real human face.

[1] Department of Materials Science and Engineering, University of Texas at Arlington, Arlington, TX, USA. [2] Department of Chemistry and Biochemistry, University of Texas at Arlington, Arlington, TX, USA. ✉email: kyum@uta.edu

Morphing thin sheets, or two-dimensional (2D) materials, into programmed 3D shapes is emerging as a paradigm in additive manufacturing[1,2]. 3D-shaped 2D materials abound in man-made and biological systems and play diverse functions[3]. Examples include flowers, leaves, organ epithelia, marine invertebrates, batoids, and bodies of vehicles. Living organisms attain a variety of morphologies of soft slender tissues and their motions through spatially controlled growth (expansion and contraction)[2–7]. Moreover, such shape-morphing materials can enable new technologies for broad applications, including soft robotics[8], deployable systems[9], microfluidics[10], tissue engineering[11], and biomimetic manufacturing[1].

Inspired by biological morphogenesis and motions, in-plane growth-induced 3D shaping approaches have shown their unique capability to create shape-changing hydrogel structures with doubly curved morphologies and motions, commonly observed in living organisms but difficult to replicate in man-made structures[2–4]. In particular, with their physical properties similar to those of biological soft tissues, such hydrogel structures have great potential for bioinspired and biomedical applications[2,12]. However, to realize the full potential of 2D growth-induced 3D shaping, theoretical and experimental challenges remain to be addressed. Fundamental among these is how to quantitatively design 2D growth for arbitrary 3D shapes and physically realize the growth in space and time to form the target shapes. For example, previous studies have mostly formed 3D shapes based on radially symmetric growth patterns or those with relatively small Gaussian curvatures, restricting the accessible space of 3D shapes[2–4]. The limitation arises from the lack of a general theoretical framework to translate arbitrary 3D shapes into 2D growth, the limited range of growth of programmable materials, and complex processes for material programming[2–4,13–16]. Moreover, forming complex shapes, such as those with both positive and negative Gaussian curvatures or multiple modular components, requires additional strategies for shape selection among multiple isometric configurations (embeddings), as a metric, or growth, prescribes local Gaussian curvatures, but not necessarily configurations, and may thus adopt multiple embeddings in space[2,4,13]. From a technological perspective, a scalable 2D material programming method capable of simultaneously fabricating multiple 3D structures with custom design could translate the shaping principle into a scalable and customizable manufacturing technology, complementing existing additive manufacturing methods[9,17,18].

Here, we show a 2D material programming approach for 3D shaping that transforms 2D materials to programmed 3D structures with doubly curved morphologies (Fig. 1, Supplementary Figs. 1 and 2). This approach prints temperature-responsive hydrogel sheets encoded with spatially controlled in-plane growth $\Omega$, which induces the shape transformations. We determine $\Omega$ for 3D shapes by conformally flattening them to the 2D plane using the boundary first flattening (BFF) method[19]. We introduce the concept of cone singularities into the design of $\Omega$ to mitigate the physical limitation of material growth, significantly increasing the accessible space of 3D shapes. As an active strategy for shape selection, we encode shape-guiding modules in $\Omega$ that direct shape morphing toward a target shape among multiple isometric configurations. Our process can simultaneously print multiple 2D hydrogels with different $\Omega$ in a single step, typically within 3 min, and on demand transform the hydrogels to programmed shapes, thereby rendering our approach scalable, customizable, and deployable. Furthermore, our flexible 2D printing process allows for sequential printing of multiple materials, enabling multimaterial 3D structures.

## Results

### 2D material programming for 3D shaping

A key challenge in growth-induced 3D shaping is to translate arbitrary 3D shapes into 2D growth. We recast this problem as a differential geometry problem of conformally mapping a target shape (surface) $M$ with metric $g$ onto the plane $C$ with metric $\tilde{g}$ (Fig. 1a, Supplementary Fig. 1)[19]. A map $f: M \to C$ is conformal if $\tilde{g} = \lambda^2 g$ for a function $\lambda^2$, where $\lambda$ is the conformal factor ($\lambda > 0$). This definition implies that a conformal mapping only allows isotropic (uniform) scaling by $\lambda$ while preserving angles, which is also the characteristic of isotropic materials (encoded with length growth $\lambda$). We thus postulated that (1) an inverse conformal mapping $f^{-1}: C \to M$ represents an in-plane growth-induced shape transformation of a 2D hydrogel $C$ to a target 3D shape $M$ (or an isometric embedding of $M$), as our hydrogels isotropically swell and shrink, and (2) $\lambda^{-1}$ quantifies the length growth (shrinkage in this study) at each point on the 2D hydrogel required to induce the shape transformation (Fig. 1a). We defined area growth as $\Omega = c\lambda^{-2}$, where $c$ is a constant. Given target 3D shapes, we computed $\Omega$ using BFF[19]. We first validated our approach by computing $\Omega$ for geometric shapes with known axisymmetric metrics: spherical cap ($K > 0$), saddle ($K < 0$), and cone ($K = 0$) with constant Gaussian curvature $K$ (Supplementary Fig. 3)[2]. The computed $\Omega$ quantitatively agree with the theoretical $\Omega$ (Supplementary Fig. 3).

We next physically realized $f^{-1}: C \to M$ with $\Omega$ using digital light 4D printing (DL4P) (Fig. 1, Supplementary Fig. 1, Supplementary Movie 1). DL4P encodes 2D hydrogels with $\Omega$ through spatial and temporal control of photopolymerization and cross-linking reactions with two types of crosslinkers with different lengths via digital light projection grayscale lithography[2]. This process increases the local density of polymer networks of hydrogels with light exposure time and thus decreases their local degrees of swelling and shrinkage, allowing for shape programming at both the swelled and shrunk states (Supplementary Figs. 4 and 5)[2]. As the thickness of a sheet vanishes, a growth-induced shape converges to an isometric embedding of a target metric[20]. The shape at the shrunk state can thus adopt a metric closer to the designed growth than one at the swelled state[2,21]. In addition, potential applications can benefit from the higher or more uniform mechanical properties of the resulting 3D structures at the shrunk state than the swelled state and the ability to form 3D shapes in physiological conditions ($T = 37\,°C$) (Supplementary Fig. 5)[2,22]. We thus formed shapes at the shrunk state ($\Omega < 1$). The resulting structures reversibly transform their shapes between $M$ and $N$ upon temperature change (Supplementary Fig. 2).

### Growth-induced 3D shaping

Figure 2 shows exemplary 3D structures created by our method. Given target shapes (Fig. 2a–d), we computed $\Omega$ (Fig. 2e–h) and printed structures using $\Omega$ (Fig. 2i–l, Supplementary Fig. 1). We first considered 3D shapes defined by a height function $h(x, y) = a(x^m + y^n)$, where $a$, $m$, and $n$ are constants (Fig. 2a, Supplementary Fig. 6). Figure 2a, i shows 3D shapes with $a = 2.09$, $m = 3$, and $n = 2$. We further formed various 3D structures, ranging from sea shell (Fig. 2j, Supplementary Fig. 7) and leaves (three leaves with the same shape but different size; Fig. 2k, Supplementary Fig. 8, Supplementary Movie 2) to automobile with open windows (Fig. 2l). Furthermore, to demonstrate a potential application of replicating real objects, we 3D scanned a model automobile and printed it (Fig. 2m). The experimentally printed structures (Fig. 2i–m) agree well with the target shapes (Fig. 2a–d, m), illustrating the ability to form structures with diverse morphologies. The shapes in Fig. 2 have a value of $\Omega_r = \Omega_{max}/\Omega_{min} < 2.15$, where $\Omega_{max}$ and $\Omega_{min}$ are the maximum and minimum values of $\Omega$, respectively. $\Omega_r$ of $\Omega$ in Fig. 2e–h, m are 2.09, 2.14, 1.63, 2.14, and 1.87, respectively, within the accessible range of growth of our material systems ($\Omega_r < 2.5$).

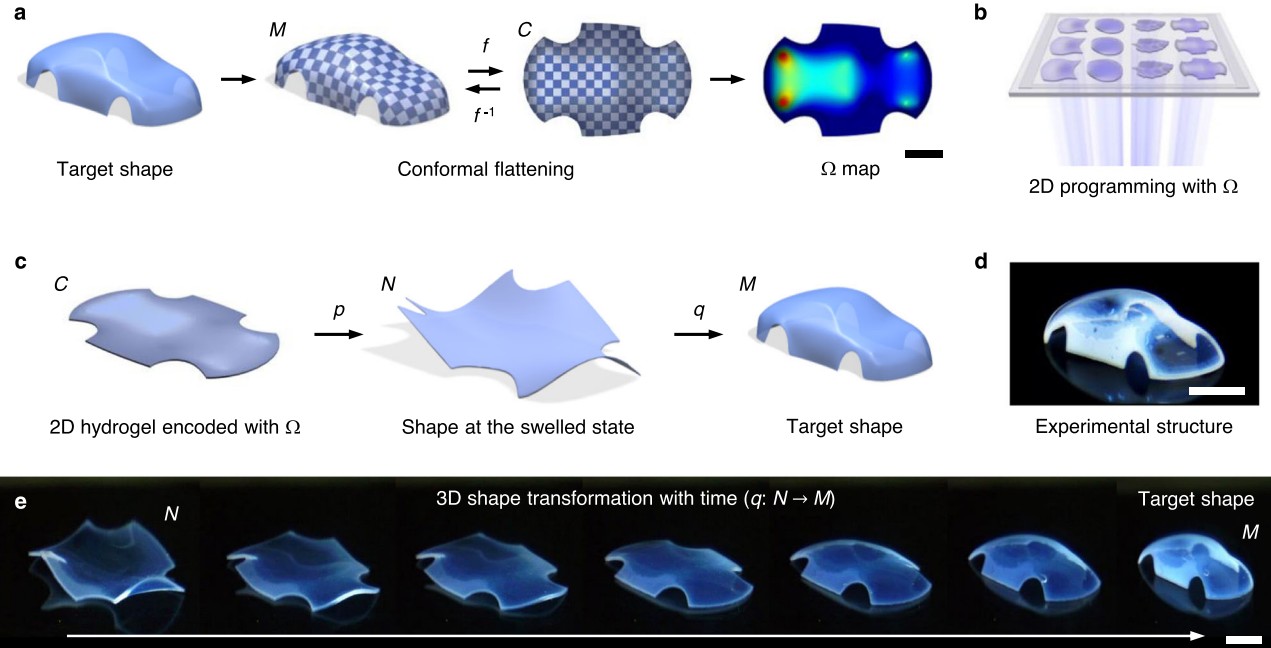

**Fig. 1 2D material programming for 3D shaping. a** Translation of a 3D target shape $M$ into 2D growth $\Omega$ via conformal mapping $f$ of $M$ to the plane $C$ ($f: M \rightarrow C$). The value of $\Omega$ ranges from 0.36 (blue) to 0.76 (red). **b** 2D printing of hydrogel sheets encoded with $\Omega$ from a single precursor solution using digital light projection grayscale lithography. **c** Schematic illustrating a shape transformation process of a 2D hydrogel $C$ encoded with $\Omega$ in **a** to the target 3D shape $M$ at the shrunk state ($f^{-1}: C \rightarrow M$). The 2D hydrogel $C$ transforms to the prescribed 3D shape $N$ at the swelled state ($p: C \rightarrow N$) upon immersion in water ($T < T_c$), where $T_c$ is the volume phase transition temperature of the hydrogel (~32.5 °C). The 3D structure $N$ at the swelled state reversibly transforms to the target 3D shape $M$ at the shrunk state ($q: N \rightarrow M$) upon temperature increase ($T > T_c$). **d** Experimental 3D structure obtained through the process shown in **a–c**. **e** Experimental 3D shape transformation of $N$ to $M$ with time. Scale bars, 5 mm in **a**; 4 mm in **d** and **e**.

**Cone singularities**. The Riemann theorem suggests that our method can form any curved shapes of thin sheets by conformally morphing 2D hydrogels[16,23]. However, in practice, the accessible $\Omega_r$ limits accessible 3D configurations[2,4]. To mitigate this limitation, we introduced the concept of cone singularities (Fig. 3). The idea is to first map a 3D shape on a cone surface ($K = 0$) and then unfold the cone surface to the 2D plane (without further area distortion) by cutting the surface through cone points, rather than directly mapping a shape to the plane (as shown in Fig. 2)[19,24]. This approach can significantly reduce $\Omega_r$, as a 3D shape can be conformally mapped to a cone surface with lower area distortion than to the plane and the cone surface can be isometrically flattened to the plane.

To demonstrate the concept of cone singularities, we first considered a hemisphere (Fig. 3a–f). A conformal mapping of a hemisphere gives $\Omega = c/[1+(r/R)^2]^2$, where $r/R$ is the normalized radial position and $R$ is a constant[2]. This mapping yields the least area distortion, as $\Omega$ is constant along the boundary[25]. Forming a full hemisphere hence requires $\Omega_r = 4$, and we can form only a portion of a hemisphere using our materials with $\Omega_r < 2.5$ (Fig. 3a). For example, the maximum attainable portion with $\Omega_r = 2.22$ is a spherical cap with a cap angle $\varphi = 70°$, where $\varphi$ is the polar angle from the pole of the cap to its base in a spherical coordinate system (Fig. 3a). Enhancing accessible $\Omega_r$ of material systems to be 4 or higher can form a full hemisphere ($\varphi = 90°$), but this approach is practically inefficient (as discussed below)[4]. We instead used a cone singularity to reduce $\Omega_r$ (Fig. 3b). Introducing a cone singularity at the pole with a cone angle $\theta_c = 60°$ (deficit angle of the cone surface) transforms $\Omega$ to have $\Omega_r = 2.22$, enabling the formation of an almost-complete hemisphere with $\varphi = 85°$ (Fig. 3a, b, Supplementary Movie 3).

To further elucidate how cone singularities transform $\Omega$, we computed $\Omega$ for a hemisphere with a cone singularity with different $\theta_c$, thus with different cone metrics, and printed the maximum attainable portion (Fig. 3c, Supplementary Figs 9–12). The maximum value of $\Omega$ and the value of $\Omega$ around the cone singularity ($r/R = 0$) decrease with $\theta_c$, decreasing $\Omega_r$ with $\theta_c$ to reach the minimum $\Omega_r = 2.22$ at $\theta_c = 60°$ (Fig. 3c, d). Further increase in $\theta_c$ leads to $\Omega < 1$ around $r/R = 0$, increasing $\Omega_r$ with $\theta_c$ at $\theta_c > 60°$ (Fig. 3c, d). Morphing a cone surface with $\theta_c < 60°$ to a spherical shape requires smaller area distortion around its tip than a flat plane, thereby decreasing $\Omega_r$ with $\theta_c$. However, as it gets sharper, a cone surface with $\theta_c > 60°$ requires larger shrinkage to smooth the sharp tip to a spherical shape, thus increasing $\Omega_r$ with $\theta_c$. The experimental structures show a quantitative agreement with the computational predictions as shown in $\varphi$ and $K$ obtained with different $\theta_c$ (Fig. 3e, f, Supplementary Figs. 10 and 12). From a practical perspective, cone singularities extend accessible $K$ for given printing areas and materials (Fig. 3f, Supplementary Fig. 12).

Increasing the number of cone singularities can further increase the accessible shape space. To demonstrate this capability, we attempted to form an almost-complete sphere (Fig. 3g–i). Directly morphing a 2D material to such a closed structure is inefficient, as it demands severe area distortion[4]. For instance, the sphere requires $\Omega_r = 234.3$, essentially impossible to achieve with available material systems, and $\Omega_r = 25.9$ with a cone singularity (Supplementary Figs. 13 and 14). To tackle this challenge, we increased the number of cone singularities to 12 and achieved $\Omega$ with $\Omega_r = 2.12$, within the accessible range of growth (Fig. 3i, Supplementary Fig. 14). Remarkably, despite its highly irregular boundary, $\Omega$ induces a sphere with well-matched cut edges, highlighting the accuracy of our approach (Fig. 3h, Supplementary Fig. 15). Beyond simple geometric shapes, the cone flattening enables us to create complex 3D structures with high $K$, difficult to attain without cone singularities. As an

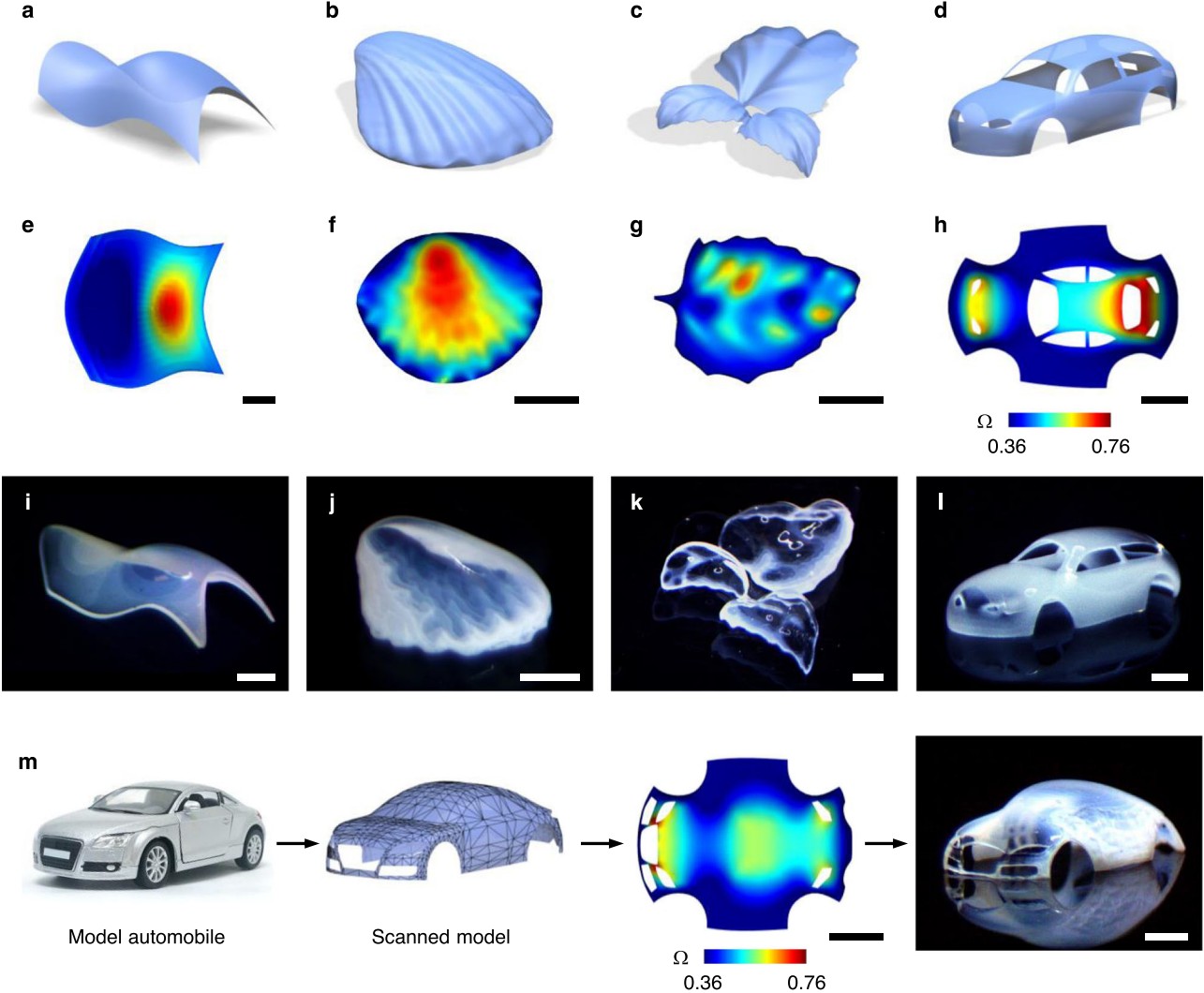

**Fig. 2 Growth-induced 3D shaping. a–d** Target 3D shapes: 3D shape defined by a height function (**a**), sea shell (**b**), leaves (**c**), and automobile with open windows (**d**). **e–h** 2D growth Ω for the target shapes in **a–d**. **i–l** Experimentally printed structures using Ω in **e–h**. We printed three leaves with the same shape but different sizes in **k**. **m** 3D scanning and printing of a model automobile. Scale bars: 5 mm in **e–h**; 3 mm in **i**; 2 mm in **j**; 5 mm in **k**; 2 mm in **l**; 5 mm (left) and 2 mm (right) in **m**.

example, we printed a nose (Fig. 3j–l). Because of high $K$ at its tip, the nose requires a large range of growth with $\Omega_r = 7.22$ (Supplementary Fig. 16). Cone flattening reduces the range to $\Omega_r = 2.06$, enabling the formation of the nose with high accuracy (Fig. 3k, l, Supplementary Fig. 16).

The use of cone singularities increases the accessible shape space, but the cuts in the resulting structures can limit their applications. A design principle that incorporates the placement of cone singularities into the function of shape-morphing structures can mitigate the limitation, which, however, requires further study[24,26]. In addition, post-printing crosslinking or surface coating processes can seal the cuts for applications that require enhanced mechanical properties.

**Shape selection**. According to Gauss's theorema egregium, Ω prescribes $K$ of a growth-induced structure but not necessarily its configuration (unless there exist constraints). In general, Ω can thus induce multiple configurations (embeddings), for example, owing to its symmetric nature[2,4]. Our method uses the slight variations in shrinkage through the thickness of hydrogels as an inherent constraint for shape selection. This constraint reduces the number of possible isometric embeddings, sufficient to adopt

target shapes for a broad range of geometries as shown in Figs. 2 and 3. This constraint breaks the up–down directional symmetry of Ω in space, directing all regions with $K > 0$ to deform in the same direction[2]. However, 3D shapes requiring complex morphing patterns, such as those with multiple $K > 0$ and $K < 0$ regions, may need additional constraints. One may predict the resulting shape by determining the embedding with minimum energy, as the metric should adopt the embedding with vanishing thickness of hydrogels[20], but actively achieving target shapes remains largely unexplored.

As an active strategy for shape selection, we encoded shape-guiding modules in Ω (Fig. 4). The modules were designed to guide the direction of shape morphing toward target shapes while minimally altering the resulting shapes. To demonstrate this approach, we printed stingray structures at two different phases of undulatory swimming motion (Fig. 4a, Supplementary Fig. 17)[27,28]. Stingrays produce complex morphologies with $K > 0$ (central body) and $K < 0$ (pectoral fins) regions, as the undulating waves travel on their pectoral fins, difficult to replicate by other methods. $\Omega_0$ (Ω for the morphology at phase 0 in Fig. 4a) and $\Omega_{0.5}$ (Ω for the morphology at phase 0.5 in Supplementary Fig. 17) induced three and two different configurations, respectively, but mostly one

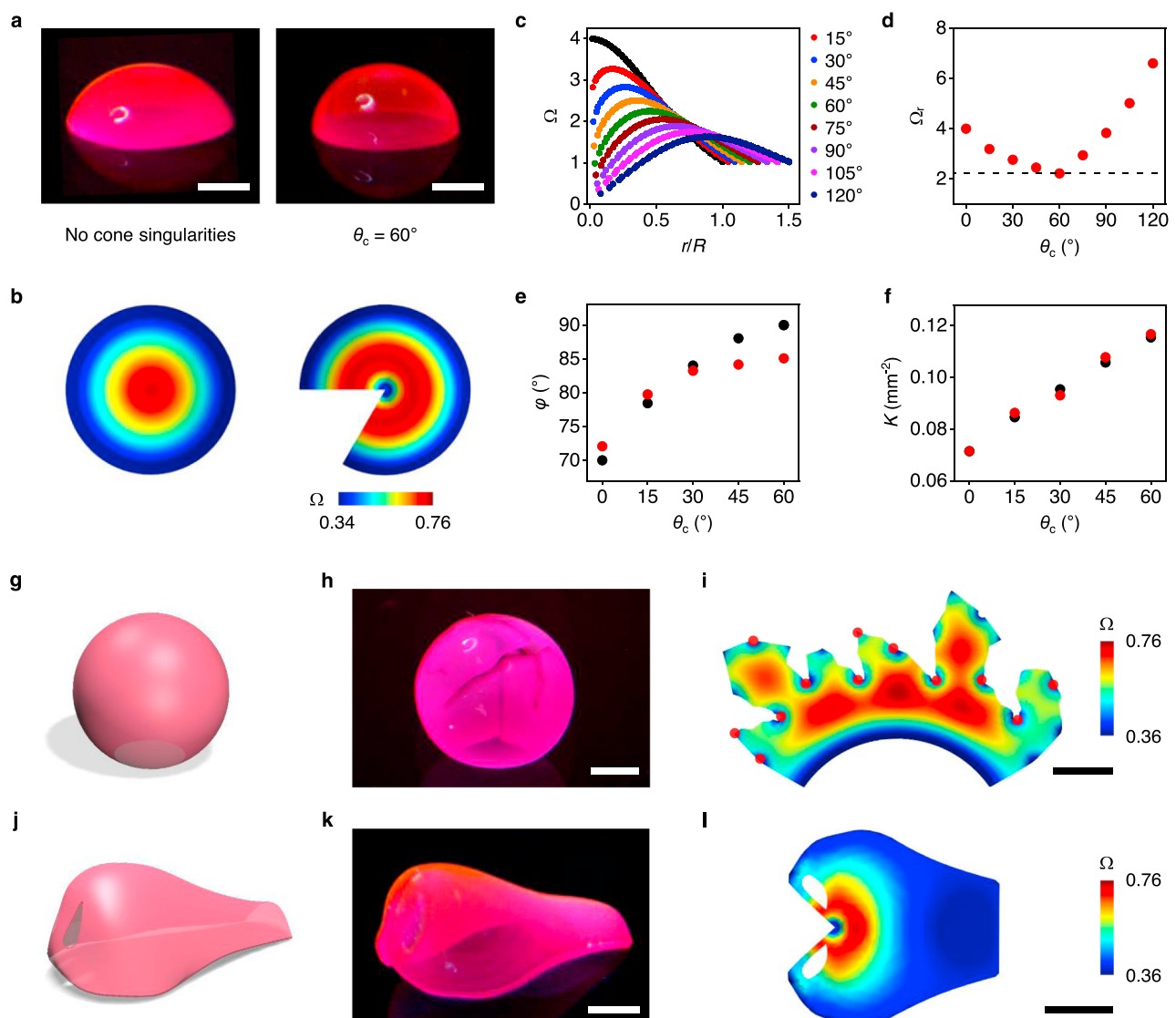

**Fig. 3 Cone singularities. a** Hemispheres formed without cone singularities (left) and with a cone singularity with cone angle $\theta_c = 60°$ (right). **b** 2D growth $\Omega$ (left) and $\Omega$ with a cone singularity with $\theta_c = 60°$ (right) used to print the hemispheres in **a**. **c** $\Omega$ computed for a hemisphere using a cone singularity with different $\theta_c$ as a function of normalized radial position $r/R$. **d** $\Omega_r$ required to form a full hemisphere with different $\theta_c$. $\Omega_r = \Omega_{max}/\Omega_{min}$, where $\Omega_{max}$ and $\Omega_{min}$ are the maximum and minimum values of $\Omega$, respectively. The dashed line indicates $\Omega_r = 2.22$. **e** Experimentally measured (red circles) and computationally calculated (black circles) cap angle $\varphi$ as a function of $\theta_c$. $\varphi$ is the polar angle from the pole of the cap to its base in a spherical coordinate system. **f** Experimentally measured (red circles) and computationally calculated (black circles) Gaussian curvature $K$ as a function of $\theta_c$. **g–i** Printing of an almost-complete sphere with 12 cone singularities: target shape (**g**), experimentally printed structure (**h**), and $\Omega$ with cone singularities (red circles) used to print the structure (**i**). **j–l** Printing of a nose with a cone singularity: target shape (**j**), experimentally printed structure (**k**), and $\Omega$ used to print the structure (**l**). Scale bars, 2 mm in **a**; 2 mm in **h**; 5 mm in **i**; 2 mm in **k**; 5 mm in **l**.

type of primary configurations (presumably the embeddings with the lowest energy), different from the target shapes (Fig. 4a, Supplementary Figs. 17 and 18, Supplementary Table 1). By contrast, $\Omega_0$ and $\Omega_{0.5}$ with a shape-guiding module with $K > 0$ in the central body yielded only the target shapes out of 16 and 36 printed structures, respectively (Fig. 4a, Supplementary Figs. 17 and 19, Supplementary Movies 4 and 5).

The shape-guiding module has two functions: (i) it directs the central body ($K > 0$) to morph in the same direction (upward or $k > 0$, where $k$ is the curvature) as the head ($K > 0$), owing to the inherent constraint (through-thickness variation in shrinkage), and (ii) it directs the pectoral fins ($K < 0$), or the principal curvatures of the fins along the direction of their major axes, to morph in the same direction as the central body along the interfaces of the body and the fins ($k > 0$), guiding the fins to

morph into the target morphologies (Fig. 4b, Supplementary Fig. 20). The symmetric nature of the metrics of the fins results in two major modes of shape morphing that the pectoral fins can select, inducing two major types of fin configurations (Fig. 4b, c, Supplementary Fig. 20, Supplementary Movie 6). The metrics of the fins and their neighboring metrics, including that of the shape-guiding module, along with inherent constraints determine the energy required for each shape-morphing mode, allowing us to direct the shape selection using shape-guiding modules. Moreover, the shape-guiding module, such as one with low $K$ in Fig. 4a and Supplementary Fig. 17, can only guide the direction of shape morphing essentially without altering the resulting shape (Supplementary Fig. 21).

Beyond 3D structures with single materials, our method can create multimaterial 3D structures (Fig. 4d). As a demonstration,

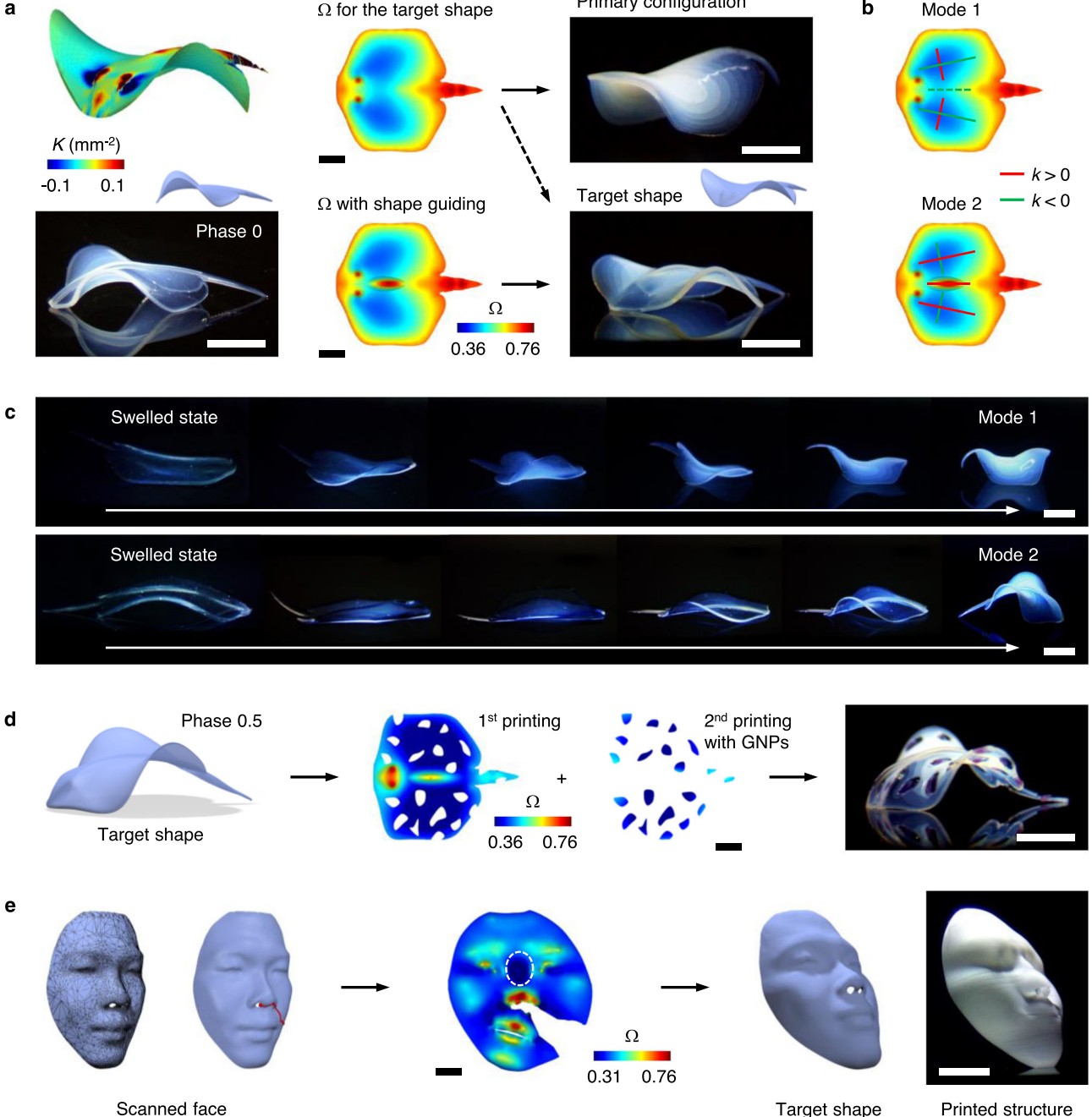

**Fig. 4 Shape selection. a** Formation of a stingray structure at phase 0 of undulatory swimming motion. The left panel shows the target shape with Gaussian curvature $K$ (top) and the side view of an experimentally printed structure (inset: side view of the target shape) (bottom). The middle panel shows 2D growth $\Omega$ computed for the target shape (top) and $\Omega$ with a shape-guiding module (bottom). The right panel shows experimentally printed structures: primary configuration induced by $\Omega$ (top) and target shape induced by $\Omega$ with the shape-guiding module (bottom). **b** Two major modes of shape morphing of the fins. The red and green lines indicate the principal curvatures and directions with $k > 0$ and $k < 0$, respectively, where $k$ is the curvature. **c** 3D shape transformation of hydrogels encoded with $\Omega$ (top) and $\Omega$ with the shape-guiding module (bottom) for a stingray structure at phase 0.5 from the swelled state to the shrunk state with time. **d** Formation of a stingray structure patterned with gold nanoparticles (GNPs) (shown in purple). **e** Replication of a real human face: reconstructed 3D shapes of a scanned face, where the red circle and line indicate a cone singularity and cutting path, respectively (left), $\Omega$ with a cone singularity computed for the face (middle), and target shape and printed structure of the face (right). Scale bars, 3 mm in **a**, **c**, and **d**; 5 mm in **e**.

we formed purple-spotted stingrays by sequentially printing the primary and gold nanoparticles (GNPs)-encapsulated hydrogels with $\Omega$ in the same layer (Fig. 4d). Furthermore, our method can print multilayer structures with single and multiple materials, highlighting its versatility (Supplementary Figs. 22 and 23). The ability to pattern functional materials in complex 3D structures

could integrate multifunctionalities into them, broadening their potential applications.

Finally, to demonstrate the ability to create truly arbitrary 3D shapes, we replicated a real human face (Fig. 4e, Supplementary Movie 7). We 3D scanned a face of a colleague, computed $\Omega$ for the face with a cone singularity, reducing $\Omega_r$ from 4.19 to 2.41,

and printed the face (Fig. 4e, Supplementary Fig. 24). In particular, the elliptical region at the nose between eyes in $\Omega$ (white dotted line in Fig. 4e) inherently functions as a shape-guiding module. This module induces an elongated saddle-like shape with $K < 0$, aligning its principal directions at the center with the major and minor axes of the module (i.e., along and perpendicular to the long axis of the nose, respectively)[2]. The shaping of the saddle directs the left and right eye sockets ($K > 0$) and the nose and forehead ($K > 0$) to morph downward ($k < 0$) and upward ($k > 0$), respectively, coordinating overall shape morphing toward the target configuration. The eye sockets and the nose and forehead morph in the opposite directions, as they share the boundaries with the elongated saddle along the directions of its major and minor axes, which deform in the opposite directions[2].

## Discussion

Inspired by biological morphogenesis and motion, we have demonstrated a 2D material programming approach to creating shape-morphing 3D structures with complex, doubly curved morphologies, often seen in living organisms but difficult to replicate with man-made materials. The ability to quantitatively design 2D growth for 3D shapes enables the creation of various 3D structures with complex morphologies. The concepts of cone singularities and shape-guiding modules further enhance the accessible space of 3D shapes and the ability to select target shapes among isometric configurations. However, our approach essentially programs 2D materials with $K$ of a target shape but not directly the shape itself, although shape-guiding modules can alleviate this limitation. An approach that programs both $K$ and mean curvature can address the limitation, including the shape selection problem, in a more general way, for example, by controlling both in-plane and through-thickness growth. In contrast to conventional additive manufacturing, which serially builds 3D structures via layer-by-layer deposition of materials, our method simultaneously prints multiple 2D materials encoded with individually customized design, without the need for printing nozzles or physical masks, and on demand morphs the 2D materials to programmed 3D structures in parallel. Our approach is thus scalable, customizable, and deployable, complementing existing additive manufacturing methods. Although this work shows the proof of concept using temperature-responsive hydrogels, the shaping principle is applicable to other programmable materials and stimuli over a range of length scales[29–31] for broad applications[1,8–11]. Our 2D printable 3D-shaping approach capable of multimaterial printing, potentially integrable with well-developed planar fabrication methods and devices, could open up new strategies for the design and fabrication of shape-morphing engineering systems, including soft robots, deployable systems, and biomimetic devices.

## Methods

**Preparation of precursors**. The precursor solution for poly(N-iso-propylacrylamide) (pNIPAm) hydrogels was prepared by dissolving N-iso-propylacrylamide (NIPAm, TCI America) (0.4 g), N,N′-methylene bisacrylamide (BIS, Sigma-Aldrich) (1 mol% of NIPAm), poly(ethylene glycol) diacrylate with an average molecular weight of ~700 g mol$^{-1}$ (Sigma-Aldrich) (0.125 mol% of NIPAm), and diphenyl(2,4,6-trimethylbenzoyl)phosphine oxide (Sigma-Aldrich) (0.15 mol% of NIPAm) in 1 mL aqueous solutions (1:3 volume ratio of water and acetone). The precursor solution for pNIPAm hydrogels with GNPs (~40–60 nm in diameters) was prepared by adding GNPs to the precursor solution for pNIPAm hydrogels (0.05 mg mL$^{-1}$). GNPs were synthesized by mixing aqueous solutions of chloroauric acid (Sigma-Aldrich) and sodium citrate (Sigma-Aldrich)[32]. GNPs were purified and concentrated by centrifugation (Centrifuge 5430, Eppendorf) before use. All materials were used as received.

**Formation of 3D structures**. Target 3D structures were formed using DL4P as described below (Fig. 1, Supplementary Fig. 1)[2]. Given target 3D shapes, growth

functions $\Omega$ were computed using BFF with custom MATLAB (MathWorks) code[19]. $\Omega$ were converted into grayscale 2D maps of light exposure time and then stereolithography (STL) files via custom MATLAB (MathWorks) code using the calibration curve of the areal shrinking ratios of pNIPAm hydrogels versus light exposure time (Supplementary Fig. 4). 2D gels encoded with $\Omega$ were prepared by polymerizing and cross-linking a precursor solution in a projection lithography cell with spatially and temporally controlled ultraviolet (UV) light using a digital light processing projector (Vivitek D912HD) with the STL files[2]. The projection lithography cell consists of a polydimethylsiloxane (PDMS) substrate, a PDMS spacer with a thickness of 400 μm (inner dimension of 57 × 32 mm), and a glass cover. For printing of large structures, fluorinated ethylene propylene film was used as a substrate and a cover to reduce the variations in the degrees of polymerization and crosslinking through the thickness of printed gels. After photopolymerization and photocrosslinking, the 2D gels were detached from the cell and immediately washed with isopropyl alcohol (IPA) and cold water (4 °C) three times each to remove uncrosslinked materials and photoinitiators and thus suppress further polymerization and crosslinking. To further remove uncrosslinked materials and photoinitiators, the gels were immersed in water at 4 °C for 16 h, exchanging the water at 1 h and 16 h after immersion. To form the target structures at the shrunk state, the temperature of the water was slowly increased to 35 °C in a temperature-controlled water bath. Food color dyes were introduced into some structures for better imaging.

**Construction of calibration curves**. Homogeneous hydrogel disks, or hydrogel disks with constant $\Omega$, with a diameter of 9 mm were prepared by uniformly exposing a precursor solution to UV light with different light exposure times of 8–70 s using DL4P as described above. To remove the possibly underexposed areas at the edges of the hydrogel disks, homogeneous hydrogel disks with a diameter of 7 mm were prepared by punching the hydrogel disks with a diameter of 9 mm using a biopsy punch (7 mm in inner diameter). The areas of the hydrogel disks with a diameter of 7 mm were measured at 35 °C. The areal shrinking ratios $A/A_0$ of the hydrogel disks with different light exposure times were used to construct the calibration curve of $A/A_0$ versus light exposure time (Supplementary Fig. 4), where $A$ and $A_0$ are the areas of the hydrogel disks at 35 °C and the areas of the hydrogel disks after punching, respectively.

**Measurement of mechanical properties**. The dynamic mechanical properties of hydrogels prepared with different light exposure times were measured using a rheometer (DHR-2, TA Instruments) with a 20-mm plate geometry. The shear storage modulus $G'$ and loss modulus $G''$ of the hydrogels at the swelled and shrunk states were measured by frequency sweeps of 0.05–10 Hz at an oscillatory strain of 1%. A Peltier plate temperature system integrated with the rheometer was used to maintain the temperature of the hydrogels at the swelled and shrunk states at 25 °C and 35 °C, respectively.

**Printing of 3D structures with multiple materials**. 3D structures patterned with GNPs were fabricated by sequentially printing primary pNIPAm and GNP-encapsulated pNIPAm hydrogels with $\Omega$ in the same layer. The primary pNIPAm gel regions were first printed using the pNIPAm precursor with $\Omega$. This printing process attaches the printed gels to the cover of the projection lithography cell. The cover of the cell with the primary gels was opened while one side of the cover was attached to the cell by tape (Scotch tape, 3M), which worked as a hinge, to align the printed gels with the second printing, and the excess precursor was removed. After washing the primary gels and the cell with IPA three times, the second precursor containing GNPs was introduced into the cell. The cover of the cell with the primary gels was then closed. The GNP-encapsulated gel regions were then printed with $\Omega$. After printing the two regions, target structures were formed as described above.

**Printing of multilayer 3D structures**. Multilayer 3D structures were fabricated by sequentially printing the first layer and the second layer using two types of spacers with different thicknesses. The first layer was printed using a first precursor with $\Omega$. This printing process attaches the printed gels to the cover of the projection lithography cell. The cover of the cell with the printed gels was opened while one side of the cover was attached to the cell by tape (Scotch tape, 3M), which worked as a hinge, to align the printed gels with the second printing, and the excess precursor was removed. The spacer of the cell (400 μm in thickness) was then replaced with a thicker spacer (typically 440–450 μm in thickness). After washing the printed gels and the cell with IPA three times, the second precursor was introduced into the cell. The cover of the cell with the printed gels was then closed. The second layer was then printed on top of the first-layer gels. After printing the two layers, target structures were formed as described above.

**3D scanning**. The model automobile in Fig. 2m was 3D scanned using a desktop 3D scanner (NextEngine) with the scanning resolution of 150 dots per inch (dpi) and dimensional accuracy of 380 μm. The face in Fig. 4e was 3D scanned using a handheld, portable 3D scanner (3D Systems) with the x–y resolution of 0.9 mm and depth resolution of 1 mm. Point cloud data created from 3D scanning were converted into triangle mesh models and smoothed. Triangle mesh models were

then used to compute Ω. The authors affirm that human research participants provided informed consent for publication of the images in Fig. 4e.

## Data availability

The data that support the findings of this study are available within the paper and its Supplementary Information and from the corresponding author upon reasonable request.

## Code availability

BFF is available in ref. [19]. Custom code is available in Supplementary Information.

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

## Acknowledgements

We thank R. Sawhney and K. Crane (Carnegie Mellon University) for discussions and assistance with BFF and H. Yang and S. Jose for discussions and technical assistance. This work was supported by the National Science Foundation (NSF) through the NSF CAREER award DMR-1848511.

## Author contributions

K.Y. conceived the research. A.N., J.J., and K.Y. designed the experiments. A.N. conducted the experiments. A.N. and K.Y. prepared the manuscript. A.N., J.J., and K.Y. contributed to the data analysis, discussed the results, and commented on the manuscript.

## Competing interests

The authors declare no competing interests.
