## [Peer Review File · Nature Communications]

REVIEWER COMMENTS

Reviewer #1 (Remarks to the Author):

In the manuscript under consideration '2D material programming for arbitrary 3D shaping', Nojoomi et al. presented the design principles on transforming conformally flattened 2D plane into target 3D shapes. A spatially varying scaling function governed the locally controlled growth, which was encoded into 2D hydrogels through tuning of polymerization degree at each pixel via grayscale image in projection lithography. Notably distinguished from previous work, the authors demonstrated in this paper a cone singularity method and shape-guiding module to relax the fabrication constraint in achievable conformal factors and resolve the non-unique configuration in shape-morphing to approach the target shape. This approach is also compatible with multilayer and multi-material situations.

However, the reviewer felt the current manuscript is premature for publication in Nature Communications, and the following major concerns should be carefully addressed in the revision:

- The very first sentence of main text implied at least part of this work was application-driven, yet the connection between examples demonstrated in the work and the potential application fields mentioned in the opening sentence was not explicitly established.
- The wording "arbitrary shape" was used repetitively in the text. Even with the cone singularity improvement to enlarge the attainable target shape space, whether or not it's actually 'arbitrary' may need more justifications. Besides, the authors claimed '... induces a seamless sphere, highlighting the accuracy of our approach' while it was apparent that the as-printed structure exhibited noticeable crevices on the surface.
- It is reasonable to assume that the utilization of soft material may compromise the structural stability of the 3D shapes, particularly when they are only consisted of thin sheets of materials. Readers might form a more complete opinion of the proposed method when its pros and cons are carefully weighed. Additionally, is the stress gradient through thickness of 2D material likely to affect the shape-morphing process? A key assumption of the conformal mapping is the isotropic swelling/shrinking of the material, how would embedded heterogeneous materials alter the isotropy of sheet-like structure considered (eg, micro fibers with a preferential alignment).
- Building on the spherical part in Fig. 3, can this method generate reentrant features? What is the maximum Gaussian curvature attainable for creating a seamless structure, given that seams may limit practicality in many applications.
- How is the prescribed, as-printed degree of crosslinking preserved for applications requiring long-term reversible morphability? It seems that all regions eventually will reach full conversion through exposure to environmental light.
- An apparent limitation of the cone singularity is introduction of cracks in the formed 3D structure, which decrease the mechanical stiffness of the structure. Could the authors comment on potential

approaches to compensate this crack problem?

- Could the authors further elaborate the relationship between the growth rate and the conversion rate (or exposure time)? Are there any considerable differences between regions with different growth rate, in terms of physical or chemical properties?
- The thickness of the hydrogel sheet is around 400 μm . If the rate of polymerization is not homogeneous in the thickness direction, how does it influence the transformation??
- In the fabrication of multilayer or multi-material structure, there is washing process between the first and second exposure. Will the washing process cause any shrinkage or distortion to the samples? How to do the alignment work?

Reviewer #2 (Remarks to the Author):

The authors describe a new concept of cone singularities to spatially control in plane growth and create designable 3D shapes by digital light printing. There are some elegant mathematical and design concepts in the publication of relevance to shape morphing and inverse design based on swelling.

The paper represents an advance in design from reference 3 (Jamal et al, 2011) where it was first reported how different low and high intensities of light (differential crosslinking) could be utilized to control curvature and swelling of a polymer and then later on how gray scale lithography could be utilized to implement this in NIPAM for 3D structures with interesting curvature (ref 9, Kim et al, 2012).

The paper is well written. However the following issues need to be addressed,

a) Since NIPAM is a thermally reversible stimuli responsive material, and a double arrow is shown in schematics, it would be good to see some complex structures put through swelling and shrinking cycles and associated reversibility/hysteresis in material programming. This data can be included in the SI for each structure or at least the exciting ones.

b) The custom code should be published in the SI so that others can see and use it. Also the limitations and scalability of the design approach should be discussed.

c) Is there a video for the assembly of the face structure. I could not find it?

d) Many more details of the models and experiments need to be provided so that the work can be reproduced and design approach used by other researchers. Apart from the code, ALL details need to be provided such as where the NIPAM, gold nanoparticles and other chemicals, materials were purchased from (since there is considerable variation from different vendors). Also details of light illumination

Point-by-point Response to the Reviewers' Comments

We thank the reviewers for their constructive and thoughtful comments. Their comments and suggestions have greatly helped us improve our manuscript. According to the reviewers' comments, we have revised our manuscript as described below.

Reviewer #1 (Remarks to the Author)

Comment 1. The very first sentence of main text implied at least part of this work was application-driven, yet the connection between examples demonstrated in the work and the potential application fields mentioned in the opening sentence was not explicitly established.

Response. We thank the reviewer for the thoughtful comment. We agree with the reviewer that this work is at least partly driven by applications, such as bioinspired manufacturing, 4D printing of soft materials and tissues, tissue engineering, and bioinspired soft robotics, but the focus of the current work is to demonstrate the 2D material programming approach for 3D shaping and its ability to create 3D structures with a variety of morphologies (potentially useful for broad applications rather than its use for specific applications). We believe that such capability could benefit broad application fields that use shape-morphing materials described in the sentence (first sentence in the original manuscript), such as soft robotics, deployable systems, (self-assembling) microfluidics, tissue engineering, and biomimetic manufacturing. We originally mentioned these application fields as examples, where shape-morphing materials can enable new technologies, to provide the general context of this work, as our 2D material programming approach for 3D shaping can potentially offer a new way to design and control the morphologies and motions of shape-morphing materials. According to the reviewer's comment, to better define the scope of this work, clarify these points, and better establish the connection between this work and its potential applications, we revised our manuscript as follows:

(1) To better define the scope of this work, we revised the first sentence in the original manuscript and move it to the end of the first paragraph (page 2): “Moreover, such shape-morphing materials can enable new technologies for broad applications, including soft robotics⁸, deployable systems⁹, microfluidics¹⁰, tissue engineering¹¹, and biomimetic manufacturing¹.”

(2) To better connect this work with hydrogels and its potential applications, we added a word “hydrogel” in the first sentence of the second paragraph of main text and added a sentence in the paragraph with Reference 2 and a new reference (Reference 12: Erol et al, Transformer hydrogels: A review. *Adv. Mater. Technol.*, 2019) (page 2): “In particular, with their physical properties similar to those of biological soft tissues, such hydrogel structures have great potential for bioinspired and biomedical applications^{2,12}.”

(3) To better connect the shaping principle in this work and the potential application fields described in the first paragraph of main text (first sentence in the original manuscript), we revised a sentence in Discussion (page 11). As such applications require different types of programmable materials or stimuli, we revised the sentence as follows: “Although this work shows the proof of concept using temperature-responsive hydrogels, the shaping principle is applicable to other programmable materials and stimuli over a range of length scales²⁹⁻³¹ for broader applications^{1,8-11}.”

Comment 2. The wording “arbitrary shape” was used repetitively in the text. Even with the cone singularity improvement to enlarge the attainable target shape space, whether or not it’s actually ‘arbitrary’ may need more justifications. Besides, the authors claimed ‘... induces a seamless sphere, highlighting the accuracy of our approach’ while it was apparent that the as-printed structure exhibited noticeable crevices on the surface.

Response. We thank the reviewer for the critical comments. We carefully thought about the use of the wording “arbitrary shape.” We agree with the reviewer that the use of “arbitrary” may need more justifications. According to the reviewer’s comment, we revised or deleted the word “arbitrary” in the revised manuscript, including the title, section titles in main text, and titles of Figs. 1 and 2.

According to the reviewer’s comment, we changed the wording “seamless sphere” to “sphere with well-matched cut edges” in page 7: “Remarkably, despite its highly irregular boundary, Ω induces a sphere with well-matched cut edges, highlighting the accuracy of our approach.”

Comment 3. It is reasonable to assume that the utilization of soft material may compromise the structural stability of the 3D shapes, particularly when they are only consisted of thin sheets of materials. Readers might form a more complete opinion of the proposed method when its pros and cons are carefully weighed. Additionally, is the stress gradient through thickness of 2D material likely to affect the shape-morphing process? A key assumption of the conformal mapping is the isotropic swelling/shrinking of the material, how would embedded heterogeneous materials alter the isotropy of sheet-like structure considered (eg, micro fibers with a preferential alignment).

Response. We thank the reviewer for the constructive comments. According to the reviewer’s comments, we revised our manuscript as follows.

(1) One of the motivations of this work with soft tissue-like hydrogels is to create shape-morphing soft materials that replicate doubly curved 3D morphologies and motions of soft tissues of living organisms, difficult to achieve with man-made materials.

To clarify the motivation and better define the scope of this work, we added “soft” in the following sentence (page 2): “Living organisms attain a variety of morphologies of soft slender tissues and their motions through spatially controlled growth (expansion and contraction)^{7-12.}”

To further clarify the motivation and better define the scope of this work, we added the following sentence with Reference 2 and a new reference (Reference 12: Erol et al, Transformer hydrogels: A review. *Adv. Mater. Technol.*, 2019) (page 2): “In particular, with their physical properties similar to those of biological soft tissues, such hydrogel structures have great potential for bioinspired and biomedical applications^{2,12.}”

As the reviewer comments, one of the limitations of hydrogels is their weak mechanical properties. We thus formed 3D shapes at the shrunk state, as pNIPAm hydrogels have enhanced mechanical properties at the shrunk state. The ability to form target 3D shapes both at the swelled and shrunk states is one of the unique advantages of our approach. We described these points in pages 4 and 5, including the following sentence. We revised the sentence that describes the mechanical properties of the hydrogels at the swelled and shrunk states with newly added data (Supplementary Fig. 5) in page 5: “Additionally, potential applications can benefit from the higher or more uniform mechanical properties of the resulting 3D structures at the shrunk state

than the swelled state and the ability to form 3D shapes in physiological conditions ($T = 37\text{ }^{\circ}\text{C}$) (Supplementary Fig. 5)^{2,22}.”

To suggest the limitation of hydrogels and the potential use of other materials, including those with better mechanical properties, we revised a sentence in Discussion (page 11): “Although this work shows the proof of concept using temperature-responsive hydrogels, the shaping principle is applicable to other programmable materials and stimuli over a range of length scales²⁹⁻³¹ for broader applications^{1,8-11}.”

(2) According to the reviewer’s comment, we discussed the key limitations of our method in the revised manuscript: (a) general limitations of our 2D material programming approach for 3D shaping and (b) limitations of the use of cone singularities.

“However, our approach essentially programs 2D materials with K of a target shape but not directly the shape itself, although shape-guiding modules can alleviate this limitation. An approach that programs both K and mean curvature can address the limitation, including the shape selection problem, in a more general way, for example, by controlling both in-plane and through-thickness growth” (page 11).

“The use of cone singularities increases the accessible shape space, but the cuts in the resulting structures can limit their applications. A design principle that incorporates the placement of cone singularities into the function of shape-morphing structures can mitigate the limitation, which, however, requires further study^{24,26}. Additionally, post-printing crosslinking or surface coating processes can seal the cuts for applications that require enhanced mechanical properties” (page 8). We also added a new reference (Reference 26: Sharp and Crane, Variational surface cutting. *ACM Trans. Graph.*, 2018).

(3) Depending on its degree, the stress gradient through the thickness of 2D materials can induce a global curvature (bending) in the resulting 3D structures, for example, as shown in Reference 10 (Jamal et al., *Nat. Commun.*, 2011). We would also like to note that as described in the first paragraph in the section of Shape selection, our method uses the slight variations in shrinkage through the thickness as an inherent constraint for shape selection. In addition, we added the discussion on the scalability of the size of 3D structures, relevant to the reviewer’s comment, in Supplementary Discussion (page 31).

(4) If heterogeneous materials (e.g., microfibers) are embedded in hydrogels with a preferential alignment, the hydrogels can show anisotropic swelling and shrinking behavior (depending on the degree of the alignment), for example, as shown in Reference 1 (Gladman et al., *Nat. Mater.* 2016) and Reference 21 (Arslan et al., *Adv. Sci.* 2019). For anisotropic materials, a new theoretical model is required to describe their shape-morphing behavior and determine the anisotropic growth required to form target 3D shapes (e.g., anisotropic metric tensor with λ_u and λ_v , which describe the growth in the parallel and perpendicular to the alignment of the embedded materials, respectively) as described in References 1 and 21.

Comment 4. Building on the spherical part in Fig. 3, can this method generate reentrant features? What is the maximum Gaussian curvature attainable for creating a seamless structure, given that seams may limit practicality in many applications.

Response. We thank the reviewer for these questions.

(1) Our method can generate reentrant features on a spherical shape. However, depending on the geometry of reentrant features, we may require shape-guiding modules as described in the section of Shape selection (Fig. 4) or transitional components described in Fig. 4 in Reference 2 (Nojoomi et al., *Nat. Commun.*, 2018). This is an important question related to the shape selection problem in the section of Shape selection (Fig. 4) or the direction control of deformation in Reference 2.

(2) The maximum accessible Gaussian curvatures K of the spherical shape (i.e., spherical cap and sphere) without ($\theta_c = 0$) and with a cone singularity are shown in Fig. 3f and Supplementary Fig. 12.

Comment 5. How is the prescribed, as-printed degree of crosslinking preserved for applications requiring long-term reversible morphability? It seems that all regions eventually will reach full conversion through exposure to environmental light.

Response. We thank the reviewer for the question. A critical step in our process is to thoroughly wash the printed gels immediately after printing to suppress further polymerization and crosslinking. To remove unreacted or uncrosslinked monomers, oligomers, crosslinkers, and photoinitiators and thus suppress further polymerization and crosslinking, we washed the printed structures with IPA and water three times each immediately after printing. To further remove unreacted or uncrosslinked monomers, oligomers, crosslinkers, and photoinitiators, we immersed the printed structures in water at 4 °C for 16 h while exchanging the water at 1 h and 16 h after immersion. In addition, as we use a UV photoinitiator with an absorbance peak at 379 nm for our printing process, polymerization and crosslinking through environment light should be negligible.

We clarified these steps in Methods of the revised manuscript (pages 13 and 14), including the following sentences: “After photopolymerization and photocrosslinking, the 2D gels were detached from the cell and immediately washed with isopropyl alcohol (IPA) and cold water (4 °C) three times each to remove uncrosslinked materials and photoinitiators and thus suppress further polymerization and crosslinking. To further remove uncrosslinked materials and photoinitiators, the gels were immersed in water at 4 °C for 16 h while exchanging the water at 1 h and 16 h after immersion.”

Comment 6. An apparent limitation of the cone singularity is introduction of cracks in the formed 3D structure, which decrease the mechanical stiffness of the structure. Could the authors comment on potential approaches to compensate this crack problem?

Response. We appreciate the reviewer’s suggestion. According to the reviewer’s comment, we provided two potential approaches that can mitigate the limitations of 3D structures formed with cone singularities for dynamic and static applications: (a) new design principle that incorporates the placement of cone singularities into the function of shape-morphing structures and (b) post-printing crosslinking or surface coating processes to seal the cuts. We added a paragraph in the section of Cone singularities with Reference 24 and a new reference (Reference 26: Sharp and Crane, Variational surface cutting. *ACM Trans. Graph.*, 2018) (page 8):

“The use of cone singularities increases the accessible shape space, but the cuts in the resulting structures can limit their applications. A design principle that incorporates the placement of cone singularities into the function of shape-morphing structures can mitigate the limitation, which,

however, requires further study^{24,26}. Additionally, post-printing crosslinking or surface coating processes can seal the cuts for applications that require enhanced mechanical properties.”

Comment 7. Could the authors further elaborate the relationship between the growth rate and the conversion rate (or exposure time)? Are there any considerable differences between regions with different growth rate, in terms of physical or chemical properties?

Response. We thank the reviewer for the constructive comments.

(1) The relationship between the areal shrinking ratio A/A_0 and light exposure time is shown in Supplementary Fig. 4. In addition, to further elaborate the mechanism that we use to control the degree of shrinkage of hydrogels by light exposure time, we added a section “Control of the degree of shrinkage of hydrogels” in Supplementary Discussion (pages 31 and 32). We added a phrase “with two types of crosslinkers with different lengths” in a sentence in page 4: “DL4P encodes 2D hydrogels with Ω through spatial and temporal control of photopolymerization and crosslinking reactions with two types of crosslinkers with different lengths via digital light projection grayscale lithography².”

(2) The regions of the resulting 3D structures at the shrunk state, formed with different light exposure times, do not have any considerable differences in physical and chemical properties. As expected from the small amount of crosslinkers, BIS (1 mol% of NIPAm) and PEGDA (0.125 mol% of NIPAm) in the precursor, the PNIPAm hydrogels, or the regions of the resulting 3D structures, formed with different exposure times are all composed mostly of PNIPAm and thus have similar chemical properties. The dual crosslinking process with two types of crosslinkers with different lengths (BIS and PEGDA) increases the density of the polymer networks of the hydrogels with light exposure time and thus decreases their degrees of swelling and shrinkage and other physical properties (e.g., Supplementary Fig. 5). However, as the hydrogels with lower densities shrink more than those with high densities, the hydrogels, or the regions of the 3D structures, formed with different light exposure times have relatively similar densities and thus similar physical properties at the shrunk state (Supplementary Fig. 5). By contrast, the hydrogels, or the regions of the 3D structures, have relatively dissimilar densities and thus dissimilar physical properties at the swelled state (Supplementary Fig. 5).

To show the physical properties of the hydrogels formed with different light exposure times, we provided Supplementary Fig. 5, which shows the density and shear storage modulus of dual-crosslinked pNIPAm hydrogels formed with different light exposure times at the swelled and shrunk states. We accordingly revised the sentence that describes the mechanical properties of the 3D structures in main text (page 5): “Additionally, potential applications can benefit from the higher or more uniform mechanical properties of the resulting 3D structures at the shrunk state than the swelled state and the ability to form 3D shapes in physiological conditions ($T = 37\text{ }^\circ\text{C}$) (Supplementary Fig. 5)^{2,22}.” We also added a section of Measurements of mechanical properties in Methods of main text (page 15).

Comment 8. The thickness of the hydrogel sheet is around 400 μm . If the rate of polymerization is not homogeneous in the thickness direction, how does it influence the transformation?

Response. We thank the reviewer for the question. If the rate of polymerization is not homogeneous in the thickness direction, the degree of shrinkage in the thickness direction can vary. The variation in shrinkage through the thickness of hydrogels can induce a global

curvature (bending or rolling) in the resulting 3D structures, depending on the degree of the variation, for example, as shown in Reference 3 (Jamal et al., *Nat. Commun.*, 2011). We would also like to note that as described in the first paragraph in the section of Shape selection, our method uses the slight variations in shrinkage through the thickness of hydrogels as an inherent constraint for shape selection.

Comment 9. In the fabrication of multilayer or multi-material structure, there is washing process between the first and second exposure. Will the washing process cause any shrinkage or distortion to the samples? How to do the alignment work?

Response. We appreciate the reviewer's comments. The washing and alignment steps are critical for the formation of multilayer and multimaterial structures. To prevent the shrinkage or swelling of printed gels and thus their distortion, we carefully washed them with IPA (rather than with water, which can induce the swelling of the gels). For the alignment, we attached one side of the cover of the projection lithography cell to the cell by tape and then opened the cover using the tape as a hinge to wash the samples and introduce the second precursor. We clarified these processes in Methods of main text in the revised manuscript (pages 15 and 16).

Reviewer #2 (Remarks to the Author)

Comment 1. Since NIPAM is a thermally reversible stimuli responsive material, and a double arrow is shown in schematics, it would be good to see some complex structures put through swelling and shrinking cycles and associated reversibility /hysteresis in material programming. This data can be included in the SI for each structure or at least the exciting ones.

Response. We thank the reviewer for the constructive suggestion. According to the reviewer's comment, we added data that show the reversibility of 3D structures: Supplementary Figs. 2, 8, and 19 and Supplementary Video 5. In addition, to reflect the reviewer's comment, we added a sentence that describes the reversible shape transformation (page 5): "The resulting structures reversibly transform their shapes between M and N upon temperature change (Supplementary Fig. 2)." We did not observe a visual difference in shape up to 4 to 5 cycles.

We would like to note that although the reversibility of the resulting 3D structures is important for their potential applications as dynamic systems, the current work mostly focuses on 2D material programming for 3D shaping (formation of target 3D shapes rather than their dynamic behavior). We thus used a single arrow from N to M (and from C to N) in Fig. 1c, which illustrate the experimental process to form a 3D target shape M , although we used a double arrow between M and C in Fig. 1a, which illustrates the theoretical process to determine 2D growth from a 3D target shape. We also used single arrows in our data that show experimental 3D shape transformations (Figs. 1e and 4c) to better define the scope of this work. As the reviewer implied, the material programming for control of the dynamic behavior of 3D structures is an interesting topic for future study.

Comment 2. The custom code should be published in the SI so that others can see and use it. Also the limitations and scalability of the design approach should be discussed.

Response. We appreciate the reviewer's thoughtful comments.

(1) We added the custom code in the SI.

(2) According to the reviewer’s comment, we discussed the key limitations of our design approach in the revised manuscript: (a) general limitations of our 2D material programming approach for 3D shaping and (b) limitations of the use of cone singularities.

“However, our approach essentially programs 2D materials with K of a target shape but not directly the shape itself, although shape-guiding modules can alleviate this limitation. An approach that programs both K and mean curvature can address the limitation, including the shape selection problem, in a more general way, for example, by controlling both in-plane and through-thickness growth” (page 11).

“The use of cone singularities increases the accessible shape space, but the cuts in the resulting structures can limit their applications. A design principle that incorporates the placement of cone singularities into the function of shape-morphing structures can mitigate the limitation, which, however, requires further study^{24,26}. Additionally, post-printing crosslinking or surface coating processes can seal the cuts for applications that require enhanced mechanical properties” (page 8). We also added a new reference (Reference 26: Sharp and Crane, Variational surface cutting, *ACM Trans. Graph.*, 2018).

(3) We provided the discussion on scalability in Supplementary Discussion (page 31): “We can rescale the size of the resulting 3D structure by uniformly rescaling the 2D printing area of Ω (e.g., Fig. 2k). However, as the size of a structure (or 2D printing area), or the ratio of the in-plane dimensions of a 2D hydrogel C to its thickness, increases, the sensitivity of shape morphing to the variations in shrinkage through the thickness increases, which can induce bending or rolling of the resulting 3D structure. For printing of large structures (e.g., face structure in Fig. 4e), we thus used fluorinated ethylene propylene film as a substrate and a cover to reduce the variations in the degrees of polymerization and crosslinking in the thickness direction of printed gels. For printing of structures larger than those in this work (e.g., 2D printing area larger than the size of our projection lithography cell with the dimension of 57×32 mm), the optimization of the thickness of hydrogels, or the ratio of the in-plane dimensions of C to its thickness, may be required.”

Comment 3. Is there a video for the assembly of the face structure. I could not find it?

Response. We thank the reviewer for the comment. We added a video for the shape formation of a face structure (Supplementary Video 7).

Comment 4. Many more details of the models and experiments need to be provided so that the work can be reproduced and design approach used by other researchers. Apart from the code, ALL details need to be provided such as where the NIPAM, gold nanoparticles and other chemicals, materials were purchased from (since there is considerable variation from different vendors). Also details of light illumination.

Response. We appreciate the reviewer’s constructive comments. As suggested by the reviewer, we provided more details of the models and experiments in the revised manuscript (highlighted in red). We also added Supplementary References.

REVIEWERS' COMMENTS

Reviewer #2 (Remarks to the Author):

The authors have satisfactorily addressed issues and the paper can be accepted.

Point-by-point Response to the Reviewers' Comments

We thank the reviewers for their constructive and thoughtful comments again. Their comments and suggestions have greatly helped us improve our manuscript.